

# Knowledge and application of sonographic scoring models for ovarian cancer management among gynecologists in Saudi Arabia: a cross-sectional study

Rana Aldahlawi

Department of Radiological Sciences, College of Applied Medical Sciences, King Saud University, Riyadh, Saudi Arabia

## ABSTRACT

**Background:** Ovarian cancer is a significant global health concern, ranking as the seventh most common cancer and the eighth leading cause of cancer-related deaths among women. Annually, it claims the lives of approximately 207,000 women worldwide. Early detection is crucial, as most cases are diagnosed at advanced stages, resulting in a 5-year survival rate of less than 20%. Common diagnostic tools include Cancer Antigen 125 (CA125) and ultrasound, but these methods are limited by sensitivity, specificity, and operator dependence. The Risk of Malignancy Index (RMI) and the Assessment of Different NEoplasias in the Adnexa (ADNEX) model, which integrates ultrasound and CA125, offer improved diagnostic accuracy. This study aims to assess the knowledge and application of these models among gynecologists in Saudi Arabia.

**Methods:** A cross-sectional study was conducted involving 148 gynecologists from various hospitals in Saudi Arabia. Participants completed a structured questionnaire that was distributed online, designed to evaluate their knowledge and application of the RMI and ADNEX models. Data were analyzed using descriptive statistics, and factors influencing the utilization of these models were identified through multivariate logistic regression analysis.

**Results:** The study found that 72% of the gynecologists were familiar with the RMI, and 58% were aware of the ADNEX model. However, only 46% reported regularly using the RMI, and 32% used the ADNEX model in their practice. Key barriers to the application of these models included a lack of training (56%), and limited access to necessary diagnostic tools (48%). Gynecologists with more than 10 years of experience were significantly more likely to use the RMI (odds ratio (OR): 2.5, 95% confidence interval (CI) [1.3–4.8]) and the ADNEX model (OR: 2.1, 95% CI [1.1–4.0]).

**Conclusion:** In Saudi Arabia, gynecologists show moderate knowledge of sonographic scoring models for ovarian cancer management, with higher familiarity for RMI than ADNEX. However, application in clinical practice is limited. Experience level influences usage, while lack of training and diagnostic access remain key barriers. Targeted educational efforts and improved resource availability are needed to support broader clinical adoption.

Corresponding author
Rana Aldahlawi,
rdahlawi@ksu.edu.sa

Sonographic scoring, Clinical decision-making

# INTRODUCTION

Ovarian cancer is a significant global health concern. It ranks as the seventh most common
cancer in women worldwide and is the eighth leading cause of cancer-related deaths
among women. According to the World Ovarian Cancer Coalition, approximately 207,000
women die from this disease each year. Early detection is crucial for improving survival
rates, as most cases are diagnosed at an advanced stage, with a 5-year survival rate of less
than 20% (*Schoutrop et al., 2022*).

Cancer Antigen 125 (CA125) is a commonly used tumor marker for assessing ovarian
cancer. However, its effectiveness is limited due to its lack of sensitivity and specificity.
CA125 levels are elevated in only 50% of patients with stage 1 ovarian cancer. Additionally,
elevated CA125 levels can occur in numerous benign conditions such as endometriosis,
adenomyosis, and pelvic inflammatory disease, as well as in several non-gynecological
conditions, including diverticulitis, liver and heart failure, and cancers of the pancreas,
breast, bladder, and liver (*Ronco, Manahan & Geisler, 2011*).

Although ultrasound is effective for detecting both benign and malignant adnexal
tumors, its accuracy depends heavily on the operator's skill and the quality of the
equipment, which introduces variability in interpretation (*Shung, 2015*). A general concern
in gynecological ultrasound is the lack of standardized terms and procedures in image
interpretation (*Timmerman et al., 2011*). Researchers have combined ultrasound with
CA125 measurements to improve the sensitivity and specificity of differentiating between
benign and malignant masses, leading to the establishment of the Risk of Malignancy
Index (RMI) (*Jacobs et al., 1990*). The Risk of Malignancy Index (RMI) is a clinical scoring
system used to estimate the likelihood that an ovarian mass is malignant. It combines three
factors: ultrasound findings (U), menopausal status (M), and serum CA-125 level.
Ultrasound is assessed based on five gray-scale features—ascites, intra-abdominal
metastases, solid areas, bilateral lesions, and multilocular lesions. A score of U = 0 is
assigned if none of these features are present, U = 1 if one feature is present, and U = 3 if
two or more are observed. Menopausal status is scored as M = 1 for premenopausal women
and M = 3 for postmenopausal women. CA-125 is measured in IU/mL and used directly in
the calculation. The RMI is computed as U × M × CA-125. Scores of 25 or less indicate low
risk, 26 to 200 indicate intermediate risk, and scores above 200 suggest a high risk of
malignancy (*Odenthal, Sharma & Fortin, 2024*). At a cut-off score of 200, the RMI has a
specificity of 97% and a sensitivity of 85%. In 1996, Tingulstad updated the RMI to RMI II,
which was recommended by the Royal College of Obstetricians and Gynecologists
(*Tingulstad et al., 1996*). However, RMI II has a sensitivity of 89% and a specificity of 73%
(*Liu et al., 2023*).

The International Ovarian Tumor Analysis (IOTA) group developed the Assessment of
Different NEoplasias in the Adnexa (ADNEX) model. It estimates the probability that an
adnexal (ovarian or surrounding) mass is benign or malignant, and further classifies

malignant tumors into subtypes. This model utilizes three clinical predictors (age, serum CA125 level, type of center) and six ultrasound predictors (lesion diameter, solid tissue proportion, cyst locules, papillary projections, acoustic shadows, and ascites) to preoperatively differentiate between benign, borderline, stage 1 invasive, stage 2–4 invasive, and secondary metastatic ovarian tumors (*Van Calster et al., 2014*). One of the advantages of the ADNEX model is its ability to calculate risk even without serum CA125 level information, although this results in a decrease in performance. The ADNEX model has demonstrated excellent discrimination between benign and malignant masses and has the potential to optimize the management of women with adnexal masses. Practitioners tasked with managing ovarian masses preoperatively must be well-versed in widely accepted models, as they facilitate precise differentiation between benign and malignant tumors, ultimately minimizing unnecessary interventions and enhancing patient care.

Several gaps exist in the literature regarding gynecologists' adoption of sonographic scoring models. Data on usage patterns are limited, especially outside specialized centers. Research on specific scoring models training adequacy and its impact on diagnostic performance is lacking. In general, the adoption of sonographic scoring models varies internationally, influenced by factors such as training availability, access to ultrasound equipment, and institutional protocols. In high-resource settings, these models are more commonly integrated into clinical practice. In contrast, low- and middle-income countries often face barriers like limited training opportunities and equipment shortages, which can hinder adoption. Additionally, the effect of these models on patient outcomes—such as referral accuracy and treatment planning—is not well documented. Few studies examine how easily these tools integrate into routine workflows or electronic systems, leaving questions about their practical usability in real-time clinical decision-making (*Buranaworathitikul et al., 2024*; *Lems et al., 2023*; *Wu et al., 2023*; *Ginsburg et al., 2023*).

Despite some gynecologists relying on subjective experience, proficient observers can consistently distinguish lesions accurately. Nonetheless, acquiring and transferring this expertise is challenging, highlighting the necessity of employing standardized models to ensure informed decision-making and improve the management of ovarian masses (*Valentin et al., 2011*). Therefore, applying morphology scoring systems such as the RMI or ADNEX model helps less experienced operators make accurate judgments in gynecology and standardizes the protocol for deciding on surgical intervention (*Viora et al., 2020*; *Lee et al., 2005*). The aim of the study is to assess how well gynecologists in Saudi Arabia know and use the Risk of Malignancy Index (RMI) and the ADNEX model for diagnosing ovarian cancer.

## MATERIALS AND METHODS

### Study design and settings

This study employed a cross-sectional prospective quantitative observational design. It was conducted at King Saud University in Riyadh, Saudi Arabia, with the survey distributed online. Data collection took place from December 2021 to December 2022. A questionnaire was spread electronically through emails and social media to the target population to ensure quick response and was completed and submitted online.

The inclusion criteria for the study encompass all physicians employed in gynecology departments within Saudi Arabia, including specialists and consultants. Professional statistical advice was sought to calculate the required sample size. We assumed a 7.5% prevalence based on findings from a recent study conducted in Saudi Arabia by Altom et al. (2024), which reported that only 29% of gynecologists had formal training in imaging studies, and 74% felt their training was insufficient. This indicates a likely low level of awareness and application of structured sonographic models, justifying our assumption of a conservative prevalence estimate. A sample size of 148 responses was determined, based on a 1.96 standard deviation, 7.5% prevalence, and 0.035 precision.

## Data collection tool

The questionnaire was written in a simple, brief format for ease of completion and consisted of three parts. The first part covers demographic characteristics such as gender, region of work, career position (such as specialist or consultant), years of experience, type of current hospital, and the number of patients seen weekly in the gynecology clinic. The second section evaluates participants' knowledge and experience using a five-point Likert scale survey. It assesses their level of training in gynecological ultrasound scanning, the availability of experienced sonographers at their institution, and their familiarity with sonographic scoring models—specifically the Risk of Malignancy Index (RMI) and the ADNEX model developed by the IOTA group. Additionally, the survey examines respondents' perspectives on the need for national validation of these diagnostic tools and the importance of aligning local practices with current international guidelines. The third part explores factors affecting surgical decision-making, considering various clinical features and diagnostic tools, the use of RMI and ADNEX models, and the influence of these scores on surgical decisions, as well as the use of other morphologic scoring systems for ovarian masses. Scores from seven Likert scale questions were combined for each respondent to calculate a total score ranging from 5 to 35. Two questions had scores ranging from 0 to 5, and the remaining five questions ranged from 1 to 5. Participants were categorized based on percentile positions: scores below the 25th percentile were classified as "Poor," those between the 25th and 75th percentiles as "Moderate," and scores equal to or above the 75th percentile as "Good."

The questionnaire was developed based on existing literature and clinical guidelines related to ovarian cancer diagnostic tools. Content validity was assessed through expert review by three senior gynecologists. Internal consistency of the knowledge and application sections was evaluated using Cronbach's alpha, which indicated acceptable reliability.

## Ethical considerations

Ethical approval was obtained from the IRB committee at King Saud University Medical City (KSUMC), approval number: E-21-6429. Completing the survey was considered as providing written consent by the respondents to participate in the study.

## Statistical analysis

The statistical analysis in this study utilized SPSS version 26.0. Cronbach's alpha was used to evaluate the validity and reliability of the knowledge and experience questionnaire. Qualitative variables were presented using frequencies and percentages, while continuous knowledge and experience scores were summarized using means and standard deviations. Fisher's exact tests were conducted to explore associations between knowledge and experience categories, socio-demographic variables, and the use of different sonographic scoring models. Significant variables from these tests were used to construct ordinal regression models, analyzing the relationship between the ordinal outcome variable and predictor variables. *P*-values less than 0.05 were considered statistically significant.

## RESULTS

Table 1 presents the validity and reliability assessment of the questionnaire used to evaluate knowledge and experience levels, comprising seven items with a Cronbach's alpha coefficient of 0.631. While slightly below the traditional threshold of 0.7, this level is considered acceptable for exploratory research involving scales with a limited number of items or multidimensional constructs (*Taber, 2018*).

The study included 148 participants, with the bigger groups from Riyadh (40.54%) and Makkah (33.11%) regions, followed by smaller groups from Tabouk, Qassim, Jawf, and Hail regions. Gender distribution showed a higher proportion of females (77.03%) compared to males (22.97%). Among the participants, the majority were gynecology consultants (60.81%), with a smaller percentage (6.76%) being gynecologic oncology consultants. Over half of the respondents (52.70%) had more than 15 years of experience in their respective fields. Specialized hospitals were the predominant workplace (43.24%), followed by secondary hospitals (37.16%) and teaching and research hospitals (19.59%). Approximately one-third of the clinicians reported seeing 10 to 20 patients weekly, and over half (53.38%) reported calculating the Risk of Malignancy Index (RMI) for their patients. Notably, a significant proportion (40.54%) reported that the total score of RMI influenced their decision for surgical intervention. Additionally, a smaller percentage (9.46%) reported using the ADNEX model, and 18.92% reported using other morphologic scoring systems. The mean knowledge scores indicated moderate levels of knowledge among participants, with scores ranging from 13.50 to 29.79 (Table 2).

The presence of specific ultrasound morphological features, such as irregular walls, cysts, septation, solid areas, papillation, or echogenicity, was prevalent in 84.46% of cases. Likewise, ascites and high tumor markers (especially CA-125) were important factors, seen in 85.14% and 83.78% of cases, respectively. The RMI and the ADNEX models were also influential, present in 75.68% and 40.54% of cases, respectively. Other factors, including MRI and CT scans, patient wishes, and general medical health, were also considered in making surgical decisions (Table 3). The majority of participants (66.89%) demonstrated a moderate level of knowledge and experience. A smaller fraction (31.76%) demonstrated good knowledge and experience, while a minimal portion (1.35%) showed poor proficiency (Fig. 1).

**Table 1 Validity and reliability of the questionnaire.**

| Knowledge and experience level assessing questions | No. of items | Cronbach's alpha based on standardized items |
|---|---|---|
| | 7 | 0.631 |

**Table 2 Demographic variables, some factors, and knowledge score of the study participants (n = 148).**

| Variables | Scale | n (%), Mean ± SD |
|---|---|---|
| Gender | Female | 114 (77.03) |
| | Male | 34 (22.97) |
| Region | Asir | 3 (2.03) |
| | Eastern Region | 13 (8.78) |
| | Hail | 1 (0.68) |
| | Jawf | 1 (0.68) |
| | Jazan | 2 (1.35) |
| | Madinah | 12 (8.11) |
| | Makkah | 49 (33.11) |
| | Qassim | 6 (4.05) |
| | Riyadh | 60 (40.54) |
| | Tabouk | 1 (0.68) |
| Position | Gynecology oncology consultant | 10 (6.76) |
| | Gynecology consultant | 90 (60.81) |
| | Specialist | 48 (32.43) |
| Experience (years) | 2–5 years | 14 (9.46) |
| | 6–10 years | 23 (15.54) |
| | 11–15 years | 33 (22.30) |
| | More than 15 years | 78 (52.70) |
| Type of hospital | Secondary hospital (150 beds or more) | 55 (37.16) |
| | Specialized hospital (Tertiary) | 64 (43.24) |
| | Teaching and Research Hospital | 29 (19.59) |
| Number of patients seen every week | Less than 10 | 13 (8.78) |
| | 10 to 20 | 49 (33.11) |
| | 21–30 | 39 (26.35) |
| | 31–40 | 47 (31.76) |
| Calculate the RMI | Yes | 79 (53.38) |
| | Some of them | 34 (22.97) |
| | No | 35 (23.65) |
| The version of RMI used | RMI I | 61 (41.22) |
| | RMI II | 20 (13.51) |
| | Other version (RMI IV) | 6 (4.05) |
| Total score of RMI affects the decision for surgical intervention | Yes | 60 (40.54) |
| | Some of them | 48 (32.43) |
| | No | 16 (10.81) |
| | Not applicable | 21 (14.19) |

| Table 2 (continued) | | |
| --- | --- | --- |
| **Variables** | **Scale** | ***n* (%), Mean ± SD** |
| Calculate ADNEX model | Yes | 14 (9.46) |
| | Some of them | 31 (20.95) |
| | No | 103 (69.59) |
| Use other morphologic scoring systems | Yes | 28 (18.92) |
| | Sometimes | 23 (15.54) |
| | Rarely | 7 (4.73) |
| | No | 90 (60.81) |
| Knowledge score | Poor | 13.50 ± 0.71 |
| | Moderate | 24.12 ± 2.84 |
| | Good | 29.79 ± 1.83 |

Note:
SD, Standard deviation; RMI, risk of malignancy index; ADNEX, assessment of different NEoplasias in the adnexa.

| Table 3 Factors affecting the decision for surgical intervention. | |
| --- | --- |
| **Factors** | ***n* (%)** |
| Overall ultrasound morphology of the ovary | 88 (59.46) |
| Ovarian diameter and size | 85 (57.43) |
| Specific features *e.g.*, irregular walls, number of cysts, septation, solid areas, papillation or echogenicity | 125 (84.46) |
| Presence of abnormal morphology bilateral | 96 (64.86) |
| Presence of doppler signal in abnormal area | 83 (56.08) |
| Presence of ascites | 126 (85.14) |
| Results of tumor marker CA-125 | 124 (83.78) |
| Other tumor markers | 86 (58.11) |
| RMI | 112 (75.68) |
| ADNEX model | 60 (40.54) |
| MRI | 101 (68.24) |
| CT | 72 (48.65) |
| Other considerations: general medical health and suitability for surgery | 84 (56.76) |
| Patient wishes | 74 (50.00) |

Note:
RMI, Risk of malignancy index; ADNEX, assessment of different neoplasias in the adnexa; MRI, magnetic resonance imaging; CT, computed tomography.

Table 4 presents the analysis of knowledge and experience levels across various socio-demographic variables and the usage of sonographic scoring models among the study respondents. The results indicate no significant differences in knowledge and experience levels based on gender ($p = 0.722$), work region ($p = 0.259$), years of experience ($p = 0.345$), and the version of the RMI used ($p = 0.374$). However, there were statistically significant differences in knowledge and experience levels based on position ($p = 0.040$), with gynecology oncology consultants having higher levels. The type of hospital also approached significance ($p = 0.053$), with specialized hospitals and teaching hospitals showing better knowledge and experience levels. The calculation of RMI ($p = 0.003$) and
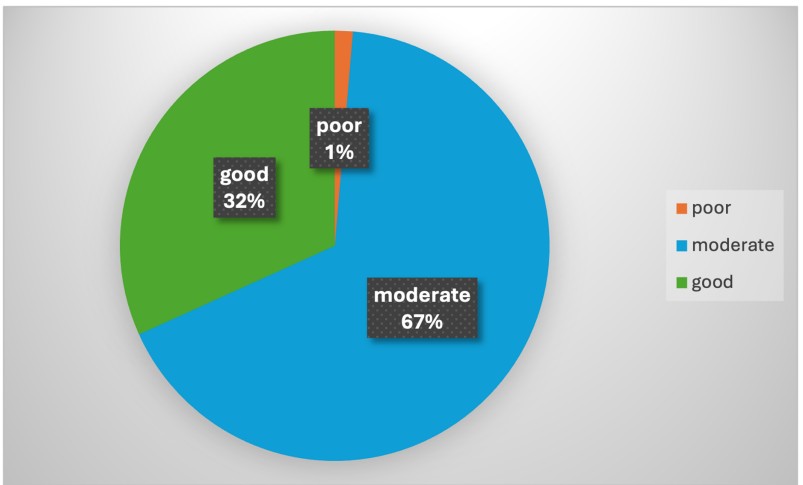

**Figure 1 Distribution of knowledge and experience levels of participants.**

**Table 4 Analysis of knowledge and experience levels across socio-demographic variables and sonographic scoring model usage among the study participants (*n* = 148).**

| Variable | Category | Poor (%) | Moderate (%) | Good (%) | *p*-value |
|---|---|---|---|---|---|
| Gender | Female | 2 (100.00) | 74 (74.75) | 38 (80.85) | 0.722 |
| | Male | 0 (0.00) | 25 (25.25) | 9 (19.15) | |
| Work region | Asir | 0 (0.00) | 2 (2.02) | 1 (2.13) | 0.259 |
| | Eastern Region | 0 (0.00) | 6 (6.06) | 7 (14.89) | |
| | Hail | 0 (0.00) | 0 (0.00) | 1 (2.13) | |
| | Jawf | 0 (0.00) | 1 (1.01) | 0 (0.00) | |
| | Jazan | 0 (0.00) | 0 (0.00) | 2 (4.26) | |
| | Madinah | 0 (0.00) | 8 (8.08) | 4 (8.51) | |
| | Makkah | 1 (50.00) | 37 (37.37) | 11 (23.40) | |
| | Qassim | 0 (0.00) | 5 (5.05) | 1 (2.13) | |
| | Riyadh | 1 (50.00) | 40 (40.40) | 19 (40.43) | |
| | Tabouk | 0 (0.00) | 0 (0.00) | 1 (2.13) | |
| Position | Gynecology oncology consultant | 0 (0.00) | 3 (3.03) | 7 (14.89) | **0.040***  |
| | Gynecology Consultant | 2 (100.00) | 65 (65.66) | 23 (48.94) | |
| | Specialist | 0 (0.00) | 31 (31.31) | 17 (36.17) | |
| Experience (years) | 11–15 years | 0 (0.00) | 22 (22.22) | 11 (23.40) | 0.345 |
| | 2–5 years | 1 (50.00) | 9 (9.09) | 4 (8.51) | |
| | 6–10 years | 0 (0.00) | 19 (19.19) | 4 (8.51) | |
| | >15 years | 1 (50.00) | 49 (49.49) | 28 (59.57) | |
| Type of hospital | Secondary hospital (150 beds or more) | 1 (50.00) | 43 (43.43) | 11 (23.40) | 0.053 |
| | Specialized hospital (tertiary) | 0 (0.00) | 39 (39.39) | 25 (53.19) | |
| | Teaching and Research Hospital | 1 (50.00) | 17 (17.17) | 11 (23.40) | |

| Variable | Category | Poor (%) | Moderate (%) | Good (%) | p-value |
|---|---|---|---|---|---|
| Number of patients seen every week | Less than 10 | 1 (50.00) | 9 (9.09) | 3 (6.38) | 0.243 |
| | 10–20 | 0 (0.00) | 37 (37.37) | 12 (25.53) | |
| | 21–30 | 0 (0.00) | 25 (25.25) | 14 (29.79) | |
| | 31–40 | 1 (50.00) | 28 (28.28) | 18 (38.30) | |
| Calculate the RMI | Yes | 1 (50.00) | 47 (47.47) | 31 (65.96) | **0.003**\* |
| | Some of them | 0 (0.00) | 21 (21.21) | 13 (27.66) | |
| | No | 1 (50.00) | 31 (31.31) | 3 (6.38) | |
| Version of RMI used | RMI I | 0 (0.00) | 39 (39.39) | 22 (46.81) | 0.374 |
| | RMI II | 0 (0.00) | 13 (13.13) | 7 (14.89) | |
| | Other version (RMI IV) | 1 (50.00) | 4 (4.04) | 1 (2.13) | |
| A total score of RMI affects the decision for surgical intervention | Yes | 0 (0.00) | 38 (38.38) | 22 (46.81) | 0.350 |
| | Some of them | 1 (50.00) | 30 (30.30) | 17 (36.17) | |
| | Not applicable | 1 (50.00) | 17 (17.17) | 3 (6.38) | |
| | No | 0 (0.00) | 10 (10.10) | 6 (12.77) | |
| Calculate ADNEX model | Yes | 0 (0.00) | 5 (5.05) | 9 (19.15) | **0.019**\* |
| | Some of them | 1 (50.00) | 12 (12.12) | 18 (38.30) | |
| | No | 1 (50.00) | 82 (82.83) | 20 (42.55) | |
| Use other morphologic scoring systems | Yes | 0 (0.00) | 18 (18.18) | 10 (21.28) | 0.417 |
| | Sometimes | 1 (50.00) | 12 (12.12) | 10 (21.28) | |
| | Rarely | 0 (0.00) | 6 (6.06) | 1 (2.13) | |
| | No | 1 (50.00) | 63 (63.64) | 26 (55.32) | |

**Notes:**
RMI, Risk of malignancy index; ADNEX, assessment of different NEoplasias in the adnexa.
\* Statistically significant.
The bold represents a statistically significant results (Such as the asterisk next to the result\*).

**Table 5 Ordinal regression analysis for knowledge and experience level and associated predictors.**

| | | Estimate (95% Confidence interval) | p-value |
|---|---|---|---|
| Knowledge and experience level = poor to moderate | | −3.534 [−5.150 to 1.918] | **<0.001**\* |
| Knowledge and experience level = moderate to good | | 2.365 [1.119–3.610] | **<0.001**\* |
| Position | Gynecology consultant | −0.498 [−1.324 to 0.327] | 0.237 |
| | Gynecology oncology consultant | 1.461 [−0.238 to 3.160] | 0.092 |
| | Specialist | Reference | |
| Calculate the RMI for the patient | Yes | 1.557 [0.345–2.769] | **0.012**\* |
| | Some of them | 1.283 [−1.106 to 2.672] | 0.07 |
| | No | Reference | |
| Calculate the ADNEX model for the patient | Yes | 1.684 [0.414–2.954] | **0.009**\* |
| | Some of them | 1.531 [0.575–2.487] | **0.002**\* |
| | No | Reference | |

**Notes:**
RMI, Risk of malignancy index; ADNEX, assessment of different NEoplasias in the adnexa.
\* Statistically significant.
The bold represents a statistically significant results (Such as the asterisk next to the result\*).

the use of the ADNEX model ($p = 0.019$) were significantly associated with higher knowledge and experience levels.

Table 5 displays the results of an ordinal regression analysis examining factors associated with different levels of knowledge and experience. The transition from poor to moderate levels of knowledge and experience was marked by a significant negative change (estimate: −3.534, $p < 0.001$). In contrast, shifting from moderate to good levels showed a significant positive change (estimate: 2.365, $p < 0.001$). Regarding professional positions, gynecology consultants did not significantly differ from specialists ($p = 0.237$). Gynecology oncology consultants exhibited a higher, though not statistically significant, level of knowledge and experience compared to specialists ($p = 0.092$). Consistent RMI calculation was significantly associated with higher levels of knowledge and experience (estimate: 1.557, $p = 0.012$). Similarly, both consistent (estimate: 1.684, $p = 0.009$) and occasional (estimate: 1.531, $p = 0.002$) use of the ADNEX model were significantly associated with higher expertise levels.

## DISCUSSION

This study provides valuable insights into the knowledge and application of sonographic scoring models among gynecologists in Saudi Arabia, specifically focusing on the RMI and the ADNEX models. The findings highlight critical areas for improvement and emphasize the importance of these scoring models in clinical decision-making for ovarian masses.

### Knowledge and application of sonographic scoring models

Our findings reveal that a significant proportion of gynecologists have moderate knowledge of sonographic scoring models. This aligns with existing literature, which also indicates variability in the knowledge and application of these models among clinicians worldwide.

Over the years, the ADNEX model has been externally validated by numerous studies (*Viora et al., 2020*; *Araujo et al., 2017*; *Chen et al., 2022*; *Epstein et al., 2016*; *He et al., 2021*; *Sayasneh et al., 2016*; *Szubert et al., 2016*), confirming its ability to discriminate between benign and malignant masses. Although, while *Van Holsbeke et al. (2009)* found that the ADNEX model was effective in distinguishing between benign and malignant ovarian masses, its utilization varied significantly across different regions and healthcare settings. This variability may be attributed to differences in training, the availability of resources, and the complexity of cases encountered in different clinical environments. Similarly, the use of the RMI has been widely adopted, yet its application is not universal, as highlighted by *Meys et al. (2017)*, suggesting discrepancies in usage based on geographic and institutional factors.

In the context of Saudi Arabia, however, local data on the validation and utility of these models remain limited. Existing studies are mostly international, and there is a need to assess how well these scoring systems perform in local populations. Differences in disease presentation, clinical infrastructure, and diagnostic resources may affect the applicability and accuracy of these tools. Without country-specific validation, clinicians may underuse or misinterpret these models, reducing their diagnostic value.

The higher knowledge levels among gynecologic oncology consultants observed in our study are consistent with findings from other research that emphasize the role of specialized training in improving the use of diagnostic tools. *Timmerman et al. (2023)* demonstrated that gynecologic oncologists are more likely to accurately interpret sonographic findings and apply scoring models effectively due to their extensive training and experience in managing complex ovarian masses. This highlights the need for continuous professional development and specialized training programs for general gynecologists to bridge the knowledge gap. Research by *Medeiros et al. (2009)* further highlights that the effectiveness of the RMI in improving diagnostic accuracy is significantly enhanced when users receive targeted training.

A critical analysis of the literature reveals that while the ADNEX model provides a more nuanced risk assessment by categorizing ovarian tumors into different histological subtypes, its adoption is still limited compared to the RMI (*Sayasneh et al., 2015*). This may be due to the more complex nature of the ADNEX model, which requires comprehensive ultrasound data and additional clinical information. In contrast, the RMI, which incorporates simpler parameters such as CA-125 levels, menopausal status, and ultrasound findings, is more straightforward to use and interpret (*Meys et al., 2017*). This simplicity likely contributes to its wider acceptance and application in various clinical settings. However, the reliance on CA-125 as a primary marker in the RMI has its limitations, particularly in premenopausal women and in cases of borderline tumors where CA-125 levels may not be elevated. As noted by *Jacobs et al. (1990)*, while CA-125 is a valuable marker, its specificity and sensitivity can be influenced by various benign conditions.

Knowledge of diagnostic models like RMI and ADNEX does not lead to consistent clinical use. Despite moderate awareness, routine adoption is low due to systemic barriers such as lack of training, limited tool access, and absence of protocols. Clinicians favor familiar methods like ultrasound morphology and CA-125. ADNEX's low use may stem from its complexity or poor workflow integration. Solutions include structured training and protocol development. Further studies are needed to explore barriers such as software availability, training adequacy, and clinician attitudes.

## Impact on clinical decision-making

The influence of the RMI and ADNEX models on clinical decisions regarding surgical intervention is evident from our results, which are supported by previous studies. The RMI and ADNEX models, by providing structured and evidence-based frameworks, significantly enhance the accuracy of diagnosing ovarian masses and guide decisions regarding surgical intervention. This is particularly critical in reducing unnecessary surgeries and improving patient outcomes. *Geomini et al. (2009)* reported that the use of the RMI significantly improved the accuracy of preoperative diagnosis, leading to better surgical planning and outcomes. Also, one large multicenter cohort study demonstrated the effectiveness of the ADNEX model in distinguishing between benign and malignant masses across all patients, regardless of whether they were managed conservatively or surgically (*Van Calster et al., 2020*). The study reported that the area under the receiver operating characteristic curve was highest for ADNEX when combined with CA125 (0.94,

95% confidence interval [0.92–0.96]), as well as for ADNEX without CA125 (0.94, [0.91–0.95]), and lowest for the RMI (0.89, [0.85–0.92]). Although not widely used, the ADNEX model was linked to higher knowledge levels in our study, showing it could be useful in clinical practice. A more recent study focused on postmenopausal women concluded that using ADNEX as a risk prediction model can improve the performance of pelvic ultrasound and effectively differentiate between benign and malignant cysts, especially for undetermined lesions (*Nohuz, De Simone & Chene, 2019*). A study by *Van Calster et al. (2014)* also highlighted that the use of the ADNEX model improves diagnostic accuracy, thereby supporting more informed clinical decisions. *Kaijser et al. (2013)* emphasized the model's effectiveness in providing detailed risk assessments that guide surgical decisions, particularly in differentiating between various types of ovarian tumors.

The study also identified key factors influencing surgical decisions, such as specific ultrasound morphological features, the presence of ascites, and tumor marker results, particularly CA-125. These factors are widely recognized in literature as critical determinants in the evaluation of ovarian masses. For instance, *Timmerman et al. (2000)* highlighted that specific ultrasound features, such as solid areas, multilocularity, and the presence of papillary projections, are strong predictors of malignancy. Similarly, the presence of ascites is a well-documented marker of advanced-stage ovarian cancer and is associated with poorer prognosis (*Jacobs et al., 1990*). Moreover, the studies have shown that the proportion of solid tissue and serum CA125 level were the strongest predictors, while the type of center was the weakest predictor, indicating that other factors were more critical in determining the malignancy rate.

Moreover, the important role of CA-125, despite its limitations, has been reaffirmed in recent studies that suggest combining it with other biomarkers and imaging findings to improve diagnostic performance. For example, *Moore et al. (2019)* demonstrated that the Risk of Ovarian Malignancy Algorithm (ROMA), which combines CA-125 with HE4, provides superior diagnostic accuracy compared to CA-125 alone. This integrated approach can potentially reduce the rates of unnecessary surgeries and ensure timely intervention for malignant cases.

We found that most clinicians prefer the RMI for surgical decisions, with the ADNEX model used less often, matching prior observations. Over half of the respondents calculate the RMI, but a significantly lower percentage utilizes the ADNEX model, reflecting broader trends. This disparity can be attributed to several factors, including the complexity of the models, as discussed earlier, as well as the availability of necessary data, and varying levels of training and confidence among practitioners. Recent advancements in training and the circulation of standardized guidelines have aimed to address these gaps. For example, educational programs that emphasize hands-on training and the practical application of these models have shown promise in enhancing clinician competence and confidence (*Medeiros et al., 2009*). Also, the integration of digital tools and online resources has facilitated access to these scoring models, potentially increasing their usage. Despite these advancements, challenges remain, particularly in ensuring that all clinicians, regardless of their location or institutional affiliation, have equal access to training and resources. This highlights the need for continued efforts to standardize education and training on these

models, as well as the importance of ongoing research to identify and address barriers to their implementation.

Recent advancements in artificial intelligence (AI) and machine learning have shown the potential to enhance the predictive accuracy of these models, offering a promising avenue for future research and development. For example, studies have demonstrated that AI algorithms can analyze complex sonographic data more efficiently and accurately than traditional methods, potentially augmenting the capabilities of existing scoring models (*Kaijser et al., 2013*). Furthermore, debates in literature highlight the need for continuous evaluation and refinement of these models. Some researchers argue for the inclusion of additional clinical variables and biomarkers to improve predictive accuracy, while others emphasize the importance of validating these models across diverse populations and clinical settings (*Sayasneh et al., 2015*). This underscores the dynamic nature of this field and the necessity for ongoing research to refine and adapt these models to evolving clinical needs and technological advancements.

The nuanced application of sonographic scoring models holds substantial potential to enhance patient management by accurately differentiating between individuals who are candidates for conservative management and those necessitating prompt surgical intervention. For instance, the ADNEX model's capability to classify ovarian tumors into distinct histological subtypes significantly enhances the precision and personalization of surgical planning, which can improve patient outcomes and reduce operative morbidity (*Jacobs et al., 1990*).

## Exploring new directions

Addressing the moderate knowledge levels among gynecologists requires innovative educational interventions to improve understanding and application of sonographic scoring models. Developing comprehensive training programs and workshops focused on the practical use and interpretation of these models could address the knowledge gaps identified in this study. Incorporating these models into standard clinical protocols and guidelines could promote consistent and evidence-based practice. To further enhance the impact and utilization of sonographic scoring models, several future research pathways can be explored. Firstly, barrier identification and overcoming strategies should be a focus. Research should delve into the barriers of adopting sonographic scoring models and identify strategies to overcome these challenges. Understanding the reasons behind the variability in model usage across different regions and healthcare settings can inform targeted interventions. Secondly, investigating the long-term outcomes of patients managed using these models could provide valuable evidence to support their routine use in clinical practice. Longitudinal studies can track patient progress and outcomes, offering insights into the effectiveness of the models over time. Thirdly, studies examining the effectiveness of educational interventions in improving knowledge and the application of these models would also be beneficial. By assessing the impact of various training programs, researchers can determine the most effective methods for enhancing clinician

competence in using sonographic scoring models. Lastly, the potential of emerging technologies, such as artificial intelligence and machine learning, to enhance the predictive accuracy of sonographic scoring models warrants exploration (*Nohuz, De Simone & Chene, 2019*). Integrating these advanced technologies can lead to the development of more sophisticated and reliable diagnostic tools.

## Strengths and limitations

This study offers valuable insights into the knowledge and application of sonographic scoring models among Saudi gynecologists, particularly focusing on the RMI and the ADNEX model. The research employed a methodologically robust approach, using structured questionnaires to gather comprehensive data from a diverse sample of practitioners across different regions. The analysis provided a nuanced understanding of factors influencing model adoption. Practical implications include recommendations for targeted training and integration of clinical guidelines, potentially impacting healthcare practices positively.

This study has several limitations. The sample may not fully represent all gynecologists practicing in Saudi Arabia, partly due to selection bias from online questionnaire distribution, which may have favored technologically inclined or academically engaged respondents. Additionally, senior consultants were overrepresented, while early-career practitioners and those in less-resourced settings were underrepresented, limiting subgroup analyses on training needs and practice variability. Self-reported data are subject to social desirability bias, especially regarding awareness and attitudes. While internal consistency was acceptable, the study lacked detailed validity metrics. The response options for non-use of models were overly broad, reducing clarity on specific barriers. The cross-sectional design limits causal inference, and the absence of contextual variables—such as hospital policies, healthcare infrastructure, and access to training—further constrains interpretation. Finally, the study did not assess actual diagnostic accuracy or patient outcomes, restricting conclusions about clinical effectiveness. Future research should address these gaps using objective data, more inclusive sampling, longitudinal designs, and refined response categories.

## CONCLUSIONS

This study highlights that gynecologists in Saudi Arabia possess moderate knowledge of sonographic scoring models for ovarian cancer management, particularly the RMI, which is more widely recognized than the ADNEX model. Despite this awareness, routine clinical application is limited. More experienced gynecologists are more likely to use these models, suggesting that clinical exposure plays a role in adoption. Key barriers include inadequate training and restricted access to necessary diagnostic tools. Addressing these gaps through structured training programs and improved resource availability is essential to support the effective integration of these models into routine gynecological practice and enhance early detection of ovarian malignancies.

## ACKNOWLEDGEMENTS

We would like to extend our gratitude to the gynecologists who participated in this study and provided invaluable insights into the diagnostic processes of ovarian masses.

### Funding

The authors received no funding for this work.

### Competing Interests

The authors declare that they have no competing interests.

### Author Contributions

- Rana Aldahlawi conceived and designed the experiments, performed the experiments, analyzed the data, prepared figures and/or tables, authored or reviewed drafts of the article, and approved the final draft.

### Human Ethics

The following information was supplied relating to ethical approvals (*i.e.*, approving body and any reference numbers):

King Saud University Medical City.

### Data Availability

The raw data is available in the Supplemental Files.

### Supplemental Information

Supplemental information for this article can be found online at http://dx.doi.org/10.7717/peerj.19746#supplemental-information.

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
