# Peer review of "Knowledge and application of sonographic scoring models for ovarian cancer management among gynecologists in Saudi Arabia: a cross-sectional study"

_PeerJ, doi:10.7717/peerj.19746_

## Round 0.1 · original submission · Major Revisions

Please be sure to thoroughly address the concerns of Reviewer 2, in a substantial revision

·

Basic reporting

There are some issues to make corrections:
1. Consent issue - How do you get verbal consent if participants receive the questionnaire by email or other social media? [Line 104 and 134]
2. Exclusion criteria - Mentioned variables are automatically excluded once the gynecologist or gynae-oncologist are included. Automatically excluded participants from outside Saudi Arabia according to the research title. [Line 110-111]
3. Recommendation- This study is to see the knowledge and practice (application) of two tools only NOT for the patient outcome!! Line [313-314]
4. Discussion - Knowledge and practice of some tools do not establish a causal relation. [Line 335]

Experimental design

This is an observation study, NOT an experimental one.

Validity of the findings

The conclusion doesn't match with the research title. It requires revision of the conclusion section. Specific comments are inserted in the REVIEWED file in REVIEW Mode. [Line 342-347]

Additional comments

None.

Reviewer 2 ·

Basic reporting

1.The manuscript has some areas where the phrasing could be clearer and more polished, which would improve its overall readability.
2.The introduction provides general background but lacks a focused discussion on the existing gaps in the literature regarding the adoption of sonographic scoring models.A more detailed justification for why this study is necessary would strengthen the rationale.
3.The manuscript includes several citations that are outdated, and the discussion relies heavily on studies published before 2020. To ensure the content remains current and relevant, it would be valuable to incorporate more recent findings, particularly those that compare the use of RMI and ADNEX in various clinical settings. This would strengthen the discussion and provide a more up-to-date perspective.

Experimental design

1. Lines 108~113: The sample size of the cross-sectional study appears relatively small. While the study mentions that 148 responses were collected based on parameters such as '1.96 standard deviation, 7.5% prevalence, and 0.035 precision,' the rationale for selecting these specific values is not clearly explained. It would be helpful to cite relevant literature or provide a power analysis to justify these calculations.
2. Lines 147~150: The validity of the measurement tool appears to be somewhat limited. The authors reported a Cronbach's alpha coefficient of 0.631, which falls below the commonly accepted threshold of 0.7. This raises some concerns about the reliability of the questionnaire.
3.The study suggests that gynecologists with more experience are more likely to use sonographic models, which is an interesting finding. However, it does not account for potential confounders such as institutional policies, access to training, or regional differences in healthcare infrastructure. To strengthen the analysis, a more rigorous statistical approach that controls for these factors would be beneficial. This would provide a clearer understanding of the relationship between experience and the use of sonographic models.
4. Lines 325~326: The authors describe the study as a 'combining structured questionnaires and interviews' but the results of the interviews were not presented or discussed. Without this information, the study currently aligns more closely with quantitative research.

Validity of the findings

1. Line 342: The study concludes that “Since ADNEX provides more detailed analysis, expanded training is needed to increase its utilization”. However, the study did not directly assess whether the training intervention was effective in promoting the application of ADNEX. The evidence provided does not establish a causal relationship.
2.The study does not compare findings to similar research conducted in other regions.Are the adoption rates of RMI and ADNEX in Saudi Arabia consistent with findings from other countries?A comparative discussion would add more depth to the interpretation of results.

Additional comments

no comment

---

## Round 0.2 · Minor Revisions

Dear authors,

Thank you for your revisions and hard-work. I'd just like a few comments / aspects clarified before acceptance:

- The recruitment through online questionnaires via social media and email could introduce selection bias favoring more technologically inclined or research-engaged practitioners. This might overestimate general awareness and knowledge levels.

- As with self-reported data, responses are subject to social desirability bias, especially concerning awareness and attitudes toward models. While Cronbach's Alpha indicates internal consistency, the actual value and content validity details are not provided. Future studies should report these metrics explicitly.
- While the data includes a high proportion of consultants and those with over 15 years of experience, the small number of other categories (e.g., fewer surgeons from smaller or less-resourced hospitals) may limit subgroup analyses, particularly relating to training needs and practice variability.

- The data shows a discrepancy between awareness and application — e.g., 72% are familiar with RMI but only 46% use it regularly. This gap is critical but seems underexplored; qualitative insights or open-ended responses could illuminate specific barriers like institutional policies, workload, or resource constraints.
- The survey assesses perceived awareness and practice but does not include actual diagnostic accuracy or patient outcomes, which could add significant value.

- The reasons for non-use (e.g., "Not applicable," "Some of them") lack detail. Incorporating more granular responses (e.g., lack of training, system issues, time constraints) would clarify modifiable barriers.

The data aligns with the manuscript’s assertion that gynecologists in Saudi Arabia have moderate awareness of RMI and ADNEX but limited routine application. The high awareness yet relatively low implementation suggest that knowledge alone does not translate into practice. This gap often stems from training deficits, resource limitations, or lack of institutional protocols. The fact that respondents cite "Overall ultrasound morphology," "CA-125," and "RMI" as key diagnostic considerations reflects awareness, but the scant use of the ADNEX model (only a minority) indicates possible barriers like complexity or unfamiliarity. Strategies to improve adoption should include structured training programs, possibly integrated into routine gynecological ultrasound curricula, and creating institutional protocols that mandate or facilitate model use. Further qualitative research could explore specific barriers—such as access to software/tools, training availability, or skepticism about models' utility—that limit widespread use.

·

Basic reporting

No comment now.

Experimental design

This is cross sectional design.

Validity of the findings

No comment.

Additional comments

The latest version is ok.

---

## Round 0.3 · accepted · Accept

Dear authors,

Thank you for your hard work and persistence. I am now accepting your manuscript for publication.